# Technology Advancements and Employees’ Qualitative Job Insecurity in the Republic of Korea: Does Training Help? Employer-Provided vs. Self-Paid Training

**DOI:** 10.3390/ijerph192114368

**Published:** 2022-11-03

**Authors:** Hyun Jung Lee, Tahira M. Probst, Andrea Bazzoli, Sunhee Lee

**Affiliations:** 1Department of Psychology, Washington State University Vancouver, Vancouver, WA 98686, USA; 2Department of Psychology, Chungnam National University, Daejeon 34134, Korea

**Keywords:** technology advancement, job insecurity, qualitative job insecurity, training, occupational health

## Abstract

While technological advancements have proliferated in our daily lives, they also pose threats to the job security of employees. Despite these growing concerns about technology-related job insecurity, little research has been carried out on the antecedents and outcomes of tech-related job insecurity. Using a cross-sectional, nationally representative survey sample of 28,989 Korean workers drawn from the Korean Working Conditions Survey, we examined the impacts of technology advancements on employee perceptions of technology-related qualitative job insecurity (i.e., perceived technology-related threat to the continued existence of valued job features) and subsequent effects on employees’ work (i.e., work engagement, job satisfaction), health (i.e., sleep), and life (i.e., work-to-family conflict) outcomes. Furthermore, we investigated the extent to which employer-provided (versus self-funded) training buffers the adverse impacts of technology advancements and associated job insecurity. The path analysis results showed more technology changes were associated with higher job insecurity, which subsequently related to adverse outcomes. While employer-provided training helped workers to reduce the negative impacts of tech changes on job insecurity, workers who paid for their training reported more adverse outcomes in face of job insecurity. We discuss these results in light of the job demands–resources theory and practical implications to buffer the adverse impacts of technology advancements.

## 1. Introduction

Technology advancements (e.g., the use of automation, computerization, and artificial intelligence (AI)) have been proliferating and changing our lives. For example, the number of organizations adopting AI has grown by 270% in four years, with more than 90% of businesses having ongoing investments in AI [1]. Although those technological advancements have brought us increased convenience and safety, many workers have been worried about their jobs or tasks being displaced by AI or robots. Indeed, according to Frey and Osborne [2], approximately 47% of jobs in the U.S. labor market (e.g., transportation and logistics, administrative support, and production occupations) are at high risk of being replaced by technology over the next 10 to 20 years. This trend appears to have been further accelerated by COVID-19 pandemic-induced lockdowns and technology adoptions by companies. For example, according to a survey during the pandemic, 43% of organizations reported that they would reduce their workforce because of technology integration [3].

The Republic of Korea is one of the countries with the highest robot density per worker, with about 55–57% of jobs at high risk of replacement [4,5]. Despite these technological changes adding potential threats to the security of one’s job itself (*quantitative job insecurity*) as well as valued aspects of one’s job (*qualitative job insecurity*), less attention has been devoted to such technological advancements as an antecedent of such insecurity [6,7]. Hence, our study attempts to shed light on the associations between technological advancement in the workplace with employee outcomes in Korea. More specifically, we aim to investigate workers’ perception of technology-related *qualitative job insecurity (Qual JI)*, which refers to perceived threats to the continued existence of important job features [8,9] in response to technological changes in workplaces. Further, we examine the outcomes of technology-related Qual JI, including work-related outcomes (i.e., work engagement, job satisfaction), health-related consequences (i.e., sleep difficulties), and the work–family interface (i.e., work-to-family conflict). Finally, because employees often seek out training to update their current knowledge and skills in the face of technology-related workplace changes, we consider the importance of distinguishing between two types of training, namely, whether the training is employer-provided or paid by the employees themselves (i.e., self-paid training). In doing so, we draw upon the job demands-resources model (JD–R) [10,11]. Figure 1 presents an overarching view of our conceptual model to be tested. Below we discuss in greater detail the theoretical and empirical foundation for the proposed relationships.

### 1.1. The Job Demands–Resources Model

The job demands–resources (JD–R) model posits that job demands and resources act together to initiate motivational as well as health-impairment processes [10,11]. Job demands are “physical, psychological, social, or organizational aspects of the job that require sustained physical and/or psychological effort”, which in turn are associated with adverse physiological and psychological outcomes [11]. As job demands require constant physical and psychological resources, workers experience a depletion of resources, ultimately resulting in adverse health outcomes (e.g., burnout) and reduced functioning (e.g., self-undermining) [10]. Based on this theorized health-impairment process, we suggest that technology changes and corresponding qualitative job insecurity operate as job demands, which can engender adverse health, work, and work–life interface outcomes. Indeed, a previous meta-analytic review showed that uncertainties associated with organizational change (e.g., layoffs, downsizing, technological automation) as a structural demand heighten Qual JI [6]. High levels of Qual JI as a stressor, in turn, have been linked to poor well-being (e.g., psychological distress, psychosomatic complaints) [12].

While job demands lead to a health-impairment process, job resources (e.g., autonomy, career opportunities) potentially assist workers to achieve work goals, stimulate personal growth and learning, and buffer the negative consequences of job demands [11]. As *employer-provided training* for upskilling provides opportunities for employees to improve their skillset and, thus, better achieve work goals, workers perceive it as a valuable resource [13]. On the other hand, *self-paid training* for upskilling often leads to skill-related learning demands falling onto workers [14]. Furthermore, self-paid training requires workers’ financial investment and time commitment outside of work hours, which put a strain on workers’ resources.

Thus, we propose that employer-provided training will serve as a valuable resource while self-paid training might act as an additional demand. As such, we suggest that employer-provided training will act as a moderator of the relationship between technology-related changes and Qual JI, such that employees who receive more training from their organizations will perceive less technology-induced job insecurity in the face of technological changes. On the other hand, given that employees who self-fund training to learn new skills are drawing from their own finite resource pool of time, money, and energies, we propose that self-paid training will exacerbate the health-impairment process, leading to a stronger relationship between technology-related Qual JI and outcomes (i.e., work engagement, job satisfaction, sleep difficulties, and work-to-family conflict).

### 1.2. Technology-Related Workplace Changes and Qualitative Job Insecurity

Artificial intelligence, robotics, algorithms, and other technology advancements have initiated workplace changes, and it has been expected that there would be a growing effect of technology advancement on workplaces [15]. As technology advancement can have a broad impact on workplaces in various ways, we define technology-related workplace changes as adoption or significant changes in information and communication devices, ways of working, and products or services. For example, many retail stores have adopted self-checkout systems to replace human employees as they are more cost-efficient [15]. Some experts have predicted that the rapid adoption of new technologies in the workplace will lead to a rise in unemployment (e.g., [16]) and heightened fears of unemployment and financial insecurity (e.g., [17]). On the other hand, others argue that AI will unlikely lead to mass unemployment but, rather, may actually create new jobs [18]. Nevertheless, it is evident that some routine *tasks*, if not jobs, will be replaced by technological innovations. For example, according to the QuantumBlack AI by McKinsey [19], companies have been increasingly adopting AI for the bottom line such as service operations optimization, product enhancement, and contact-center automation. The report further stated that the adoption of AI has been beneficial to companies in terms of cost savings. As such, the trend of a computer replacing certain bottom-line tasks is expected to be continued. Hence, although the impact of technology-related organizational changes on *quantitative* JI might not be clear, we expect that technology advancements result in changes to important job features, threatening the established positions of many middle- and lower-class workers.

As previously mentioned, organizational changes due to technological advancement, as a structural demand, heighten uncertainty and unpredictability of the workforce in the organizations, thereby workers perceive job insecurity [6,20,21,22]. Moreover, technology-related changes increase workers’ perceptions of being less-skilled by making workers’ current skillsets outdated. For example, hotel employees expressed mixed feelings in response to AI adoption; while it helped to eliminate human errors, it could also take away major tasks from human workers [7]. As such, technology-related changes are predicted to engender fear of losing important job features. Hence:

**Hypothesis 1.** *Technology-related workplace changes will be positively related to technology-related Qual JI*.

### 1.3. Technology-Related Qualitative Job Insecurity and Employee Outcomes

Individuals who worry about their professional growth due to the fear of losing essential job features experience increased strain and reduced motivation, thus leading to adverse work and health outcomes. Previous studies support these work- and health-related adverse impacts: Qual JI was related to employees’ negative attitudinal responses such as increased turnover intention, as well as decreased job satisfaction, work engagement, and organizational commitment [8,23,24]. Similarly, it has been reported that workers who perceive threats of losing important job features experience significant physical and mental health complications including psychological distress, psychosomatic complaints, and emotional exhaustion [8,12,23,24].

In addition to job- and health-related outcomes of Qual JI, recent studies have suggested adverse effects of Qual JI on work-to-family conflict [25]. Work–family conflict refers to the inter-role conflict between work and family as the multiple role demands require resources and are mutually incompatible in some way [26]. In particular, we suggest that technology-related Qual JI drain requires workers’ psychological and physiological efforts in work domains and inhibits workers from successfully fulfilling their family roles, resulting in work-to-family conflict.

As technology-related Qual JI is a specific type of Qual JI, it is plausible to expect similar work-, health-, and work–family outcomes in response to the heightened technology-related Qual JI. Thus, drawing upon JD–R theory and the empirical evidence reviewed above, we expect that:

**Hypothesis 2.** *Technology-related Qual JI will be related to more adverse outcomes, specifically: (a) decreased work attitudes (i.e., less work engagement, job satisfaction); (b) impaired health (i.e., more sleep difficulties); and (c) more work-to-family conflict*.

### 1.4. Employer-Provided vs. Self-Paid Training

As technological advancements have affected production processes, business models, and the way people communicate [27], demands for high communicative and cognitive skills, creativity, social intelligence, and comprehension have increased [28,29]. With these changes, reskilling or upskilling to adapt to these advancements has become crucial for workers. Indeed, the World Economic Forum reported that 50% of all employees will need reskilling by 2025, while 40% of employees’ core skills are expected to change in the next five years [3]. Academic research also indicates that technology advancement intensifies an individual’s perception of skill-related learning demands as workers are often expected to adjust their skills in response to technology-related changes [14,30]. As such, training opportunities for reskilling/upskilling serve as a crucial resource for workers to adapt to new technological developments. We propose that whether such training is provided by one’s employer versus self-funded will impact the relationships between technology change, Qual JI, and the posited work, health, and life outcomes.

JD–R theory posits that increased job resources buffer the adverse impacts of job demands (such as technology advancement at workplaces) on strain [10]. We suggest that *employer-provided* training for reskilling/upskilling act as valuable resources to buffer the relationship between technology-related changes and technology-related Qual JI. First, employer-provided training reduces the gap between workers’ current vs. new skillsets that are required to work with computerization, thereby assisting workers to reduce the perceptions of skill-related learning demands and its negative impacts on the perception of losing essential job features. Second, employees who are provided training for upskilling/reskilling perceive that their employers invest resources in the employees, thus having an increased level of perceived organizational support (i.e., the perception that organizations value employees’ contributions and care about their well-being) in times of technological changes [31,32]. As such, employer-provided training provides constructive (i.e., skills, knowledge, perspectives, experiences) and social resources (i.e., organizational support) [33]. Both of these two types of resources have been empirically supported to reduce employees’ Qual JI [6]. Based on these, we hypothesize that:

**Hypothesis 3.** 
*Employer-provided training will buffer the relationship between technology-related workplace changes and technology-related Qual JI.*


Contrary to employer-provided training, *self-paid* training might not serve as a resource, but rather a coping strategy against learning demands. As suggested above, employees have heightened skill-related learning demands in response to technology changes and corresponding Qual JI. When employees perceive that employer-provided training is not sufficient to develop required skills for their current jobs or they do not receive proper training from their organization, employees might choose to self-fund the training necessary to deal with Qual JI. Indeed, Demerouti and colleagues suggested that individuals are not passive recipients of unfavorable external influences (i.e., technology-related Qual JI); rather, they are active modifiers in that they develop individual strategies via intentional behaviors such as coping [34]. However, coping as a response to high demands tends to be less effective [10]. As such, we speculate that for those workers who choose to educate themselves in response to felt Qual JI, self-paid training might not be an effective option. Rather, since workers who initiate self-paid upskilling need to invest their own resources (e.g., money, time, energy) into it, we expect they may experience a drain of valuable resources when they are already threatened with a loss of important job features. For example, workers need to recover from work during nonwork hours in order to replenish their energy back to the pre-stressor levels [35]. However, work-related nonwork time activities (i.e., self-paid training) would further prevent employees’ recovery (e.g., psychological detachment from work, relaxation), when they are already experiencing rumination because of Qual JI [36,37]. Thus, self-paid training could intensify the adverse impact of technology-related Qual JI on work and health outcomes. Therefore, we hypothesize that:

**Hypothesis 4.** 
*Self-paid training will strengthen the relationship of technology-related Qual JI with (a) work attitudes (i.e., work engagement, job satisfaction), (b) impaired health (i.e., sleep difficulties) and (c) work-to-family conflict.*


## 2. Method

### 2.1. Participants and Procedure

The Republic of Korea has the highest robot density per workers (i.e., 932 robots installed per 10,000 employees) in the world, which is seven times higher than the global average of 126 [5]. Additionally, the Korean government recently announced its continued commitment to adopting artificial intelligence and robots. As such, it is incumbent to study the impact of technology advancements for employees in the context of Korea. Furthermore, to our knowledge, the 6th Korean Working Conditions Survey (KWCS) is the first nationally represented survey data that measured technology related Qual JI.

We tested our hypotheses using a cross-sectional dataset of 28,989 workers in Korea collected in 2021. The dataset is part of the 6th KWCS, which uses multistage probability proportion stratified cluster sampling to investigate various working conditions related to workers’ safety and health through a combination of computer-assisted face-to-face interviews, internet, and paper-and-pencil surveys with nationally representative workers aged 15 years or older. The survey was a nationally approved by Statistics Korea and was operated in accordance with Article 18 of the Statistics Act (approval No: 380002). To ensure the quality of data, quality control fieldwork was conducted for the entire sample using a structured interview script and the cases in which errors such as false entries were found were excluded from the dataset and replaced with new interviews [38].

The original dataset included 50,538 respondents, but we excluded those who were older than 65 years old (*n* = 5295), self-employed (*n* = 8374), and unpaid-family workers (*n* = 1429) as we were interested in the impact of technology-related workplace changes and the role of employer provided training for paid employees. Sixty-five years old was used as a cut-off point as those who are older than 65 are protected by the welfare of senior citizens act. Further, those who had missing values (*n* = 6451) were excluded from the analysis. The vast majority were permanent workers (81.9%), with average working hours of 46.67 per week (*SD* = 11.81). About half of the respondents were female (52.3%) and graduated college or higher (58.9%), with a mean age of 43.54 (*SD* = 11.73). The three most represented industries were manufacturing (18.93%), wholesale and retail (15.01%), and health care and social welfare services (11.12%). Finally, office (24.42%), professional/semi-professional (22.27%), service (12.36%), and sales workers (12.13%) were the most represented occupations. Table 1 shows sociodemographic information for the sample.

### 2.2. Measures

The scales used in the 6th KWCS were adopted from the 2020 European Working Conditions Survey (EWCS) and translated into Korean [39]. A panel of 6 subject matter experts reviewed the questionnaire to ensure content validity and the cognitive interview as well as pilot study were conducted to make sure the quality of the questionnaire [38]. Items for technology-related changes and technology-related Qual JI were newly developed for the 2020 EWCS and other scales were used in the previous waves of EWCS [39].

*Technology-related changes* in the workplace were assessed using the average of three items regarding the new introduction of or significant changes in technology-related: (1) information and communication devices, (2) ways of working, and (3) products or services for the past three years (0 = No, 1 = Yes).

To measure *technology-related Qual JI*, respondents were asked the extent to which they were concerned about the impacts of technological advances and automation on each of the five features (e.g., “Future changes to your job that may make it more difficult to use your skills and abilities”, α = 0.90). Items were asked on a 4-point scale (1: very worried–4: not at all worried) and reverse-coded so that higher scores reflect higher job insecurity.

Respondents were also asked if they have received any *training* to improve their task-related skills or work performance during the past year, including both: (1) training paid or provided by their employer and (2) self-paid training (0 = No, 1= Yes).

Finally, *work engagement* was measured using a 3-item scale (α = 0.80; sample item = “At my work, I feel bursting with energy”) with a response scale ranging from 1 (always) to 5 (never). *Job satisfaction* was measured with a single item (“Overall, are you satisfied with your working conditions in your job?”) on a range from 1 (very satisfied) to 4 (not at all satisfied). *Sleep difficulties* was assessed with a 3-item scale (α = 0.87; sample item = “having difficulty to get to sleep”) with a response scale ranging from 1 (daily) to 5 (never). Finally, *work-to-family conflict* was assessed with 3 items (α = 0.80; sample item = “My work prevents me spending sufficient time with my family”) with a response scale ranging from 1 (always) to 5 (never). Responses for these four scales were all reverse-coded such that higher scores reflect higher levels of work engagement, job satisfaction, sleep difficulties, and work-to-family conflict, respectively.

### 2.3. Data Analysis

Before testing our structural hypotheses, skewness and kurtosis of the study variables were assessed to check if our model satisfied normality assumptions. Our preliminary checks showed that skewness and kurtosis of most variables showed no violation except for the self-paid training variable (skewness = 4.73, kurtosis = 20.38). Thus, we used maximum-likelihood estimation with robust standard error (MLR) for our analysis as implemented by Mplus 8.8 [40]. The goodness of fit of the measurement model with five latent variables (i.e., technology-related Qual JI, work engagement, job satisfaction, sleep difficulties, and work-to-family conflict) to the data was checked. The measurement model showed a good fit [41]: χ^2^ (71) = 5438.41, CFI = 0.97, RMSEA = 0.05 (90% Confidence Interval = [0.050, 0.052]), SRMR = 0.02. Finally, we tested the hypothesized path model as shown in Figure 1.

## 3. Results

The study variables’ mean, standard deviation, and correlations are shown in Table 2. As expected, employees reporting more technology-related changes indicated higher levels of tech-related Qual JI (*r* = 0.49, *p* < 0.001). Similarly, more technology-related Qual JI was also associated with lower job satisfaction (*r* = −0.07, *p* < 0.001), more sleep difficulties (*r* = 0.10, *p* < 0.001), and greater work-to-family conflict (*r* = 0.16, *p* < 0.001). However, it was not significantly related with work engagement (*r* = 0.00, *p* = 0.72). Finally, employees reporting more tech-related job changes also reported more employer-provided (*r* = 0.22, *p* < 0.001) and self-funded (*r* = 0.11, *p* < 0.001) training.

The structural model showed a modest fit [41]: χ^2^ (10) = 2012.98, CFI = 0.82, RMSEA = 0.08 (90% Confidence Interval = [0.080, 0.086]), SRMR = 0.04. As shown in Table 3, technology-related changes were positively related to technology-related Qual JI (*b* = 0.19, *SE* = 0.02), supporting H1. Interestingly, the direct effect of tech-related changes was positively related to levels of job engagement (*b* = 0.17, *SE* = 0.01) and job satisfaction (*b* = 0.10, *SE* = 0.01). However, more changes were also associated with more sleep difficulties (*b* = 0.23, *SE* = 0.02) and greater work-to-family conflict (*b* = 0.26, *SE* = 0.02). Next, in partial support of H2, higher levels of technology-related Qual JI were associated with less job satisfaction (*b* = −0.05, *SE* = 0.00), more sleep difficulties (*b* = 0.10, *SE* = 0.01), and greater work-to-family conflict (*b* = 0.17, *SE* = 0.01), but not significantly related to work engagement (*b* = 0.004, *SE* = 0.01). Thus, H2a was partially supported while Hypotheses 2b and 2c were fully supported.

As we hypothesized in H3a, employer-provided training buffered the relationship between technology-related changes and technology-related Qual JI (interaction term = −0.12, *SE* = 0.03). In other words, the positive association between technology-related changes and technology-related Qual JI was reduced when the employer provided their employees with training. On the other hand, self-paid training strengthened the adverse impact of technology-related Qual JI on work engagement (interaction term = −0.13, *SE* = 0.03), sleep difficulties (interaction term = 0.12, *SE* = 0.3), and work-to-family conflict (interaction term = 0.11, *SE* = 0.03), but not job satisfaction (interaction term = −0.04, *SE* = 0.02). In other words, the effects of Qual JI on work engagement, sleep problems, and work-to-family conflict were exacerbated when employees self-funded their upskilling. Thus, H4 was partially supported.

We also examined mediation effects as a function of training status (see Table 4): the adverse indirect impacts of technology-related changes on work engagement, job satisfaction, sleep difficulties, and work-to-family conflict through technology-related Qual JI were significant (*p* < 0.01) when employees did not receive any training from their employers but paid for their own training. For other combinations of employer-provided training and self-paid training cases, the results were not consistent across different outcomes. Finally, the total effects of technology-related changes (i.e., direct plus indirect effects) on outcomes were significant for all outcomes regardless of training status.

## 4. Discussion

### 4.1. Summary of Results

Accelerated by the COVID-19 pandemic, the world of work is undergoing momentous change. Nearly half of all occupations (including those in production, transportation, extraction, agriculture, and maintenance/repair) are at risk of being automated within the next two decades [2]. Not surprisingly, estimates suggest that approximately half of all employees will require some level of reskilling and upskilling by 2025 due to technological advancements in the workplace such as cloud computing, reliance on big data, e-commerce, and the incorporation of robotic and artificial intelligence technology [3]. At the same time, research indicates that 85 million jobs may be displaced by the changing nature of the roles played by humans and machines [3].

Given these momentous shifts, the purpose of our study was to empirically assess whether such technology-induced changes in the workplace prompt the perception of greater qualitative job insecurity (i.e., threats to valued job features) and, subsequently, more adverse work, health, and life outcomes. Moreover, given the demands for upskilling and reskilling, it is critical to determine who should initiate and be responsible for such training provision: the employee or the employer. Therefore, we additionally examined the impact of employer- vs. self-funded training on the proposed relationships between tech-related changes at work, experienced Qual JI, and important work, life, and health outcomes.

As expected, among a sample of Korean workers (who face some of the highest levels of automation in the world), we found a positive association between tech-related changes and employee reports of qualitative job insecurity. In other words, when faced with more technology-induced changes at work, employees indicated more worries that valued features of their job were threatened or at risk. Moreover, employees who perceived higher Qual JI reported that they were less satisfied with their job, experienced more difficulties sleeping, and had more inter-role conflict between their work and family domains (specifically work interfering with their family role).

Interestingly, tech-related changes themselves did not have uniformly negative effects. Indeed, more changes were associated with greater work engagement and job satisfaction, suggesting that there can be beneficial effects due to the adoption of new technology. Moreover, as predicted when employers supported their workforce by providing training opportunities, this support helped to buffer the otherwise negative impact of tech-related changes on tech-related Qual JI. Thus, it appears that employers can short-circuit the often-adverse downstream effects on employee work, family, and health outcomes.

On the other hand, when employees take on the added burden of seeking out upskilling or reskilling opportunities, this appears to magnify the damaging effects of technology-related Qual JI. Indeed, among employees who paid for additional training out of pocket, Qual JI was associated with worse work engagement, more sleep difficulties, and higher levels of work-to-family conflict compared to employees who did not expend these additional personal time, money, and energy resources.

### 4.2. Theoretical and Practical Implications

From a theoretical perspective, our findings lend further support for the validity of the job demands–resources model [11]. The JD–R model posits that demands placed on workers can trigger a health impairment process, whereas the provision of resources can instigate motivational processes. Our results indicate that taking on the time, energy, and monetary burden of self-funding one’s skills portfolio not only negatively impacted health outcomes (i.e., sleep disturbances), but also magnified the adverse effects on work engagement and work-to-family conflict in response to Qual JI. Thus, while the JD–R model focuses on health impairment, the model may also be applicable to other forms of impairment within work and life domains. On the other hand, our results clearly supported the theoretically expected beneficial effects of employer-provided training programs. Indeed, among employees who were provided such training, the relationship between encountering tech-related changes and felt Qual JI was significantly attenuated.

From a practical standpoint, our results highlight the important role played by employers in helping today’s workforce successfully manage rapidly changing technology advancements and innovations. While employers might argue that employees should be free agents in charge of their own career development, relying on employees to self-fund their upskilling and reskilling in the face of Qual JI appears to have deleterious effects for employees and employers alike in terms of work engagement, employee health, and work/nonwork conflicts, particularly when they are faced with Qual JI induced by technology changes at work. Thus, it is in an employer’s self-interest to not only have a workforce with the latest skills but also fund the investment in employees needed to make that happen. Without this, the qualitative job insecurity otherwise experienced by employees when faced with technological change at work can be expected to replicate prior research on the adverse effects on work, health, and family outcomes (e.g., [8,12,23,24,41]).

### 4.3. Limitations and Directions for Future Research

While this represents the first large-scale and nationally representative investigation of the impacts of technology-related changes in the workplace on perceived Qual JI, as well as the moderating influence of employer- vs. self-funded training, the data are nonetheless cross-sectional in nature. As such, it is not ideal to test hypothesized mediation effects, and therefore, future research is needed using multiwave data in order to more rigorously test these propositions. Similarly, while the nature of the Korean Working Conditions Survey allows one to draw generalizable inferences to the broader working population in South Korea, future research is needed to evaluate whether these findings apply to other countries and cultures, particularly those where the incorporation of robots and automation is not as advanced.

Future research should also endeavor to capture a more comprehensive assessment of the variables of interest, particularly with respect to the nature of the self- vs. employer-funded training. Because the KWCS covers a wide array of topics, capturing data from 50,000 individuals using computer-assisted face-to-face interviews, many of the measures are necessarily very brief or even single-item measures with unknown reliability and validity. For example, during the course of the interview, respondents were asked to indicate if they had received any training to improve their task-related skills or work performance during the past year that had either been a) paid or provided by their employer or b) self-funded. While the dichotomous responses (yes/no) for each type of training allowed us to test our moderating hypotheses, future research should capture additional detail regarding the content and quality of such training (e.g., was the training specifically to address technology changes in the workplace?).

In a similar fashion, our current investigation was limited by the existing KWCS content and therefore could only evaluate a subset of the myriad of outcomes known to be impacted by the experience of job insecurity. Therefore, future research should also evaluate whether the currently observed relationships hold for other outcomes such as in-role task performance, creativity and innovation, and extra-role performance to determine whether these impacts on these variables might also be attenuated by the provision of employer-provided upskilling opportunities. Additionally, although the main effects of technology changes on outcomes were not our main interest, our results suggested that technology changes have direct beneficial effects on job attitudes while showing adverse effects on sleep and work-to-family conflict. As such, it might be interesting to examine if these varying consequences of technology changes on other work- vs. health- outcomes hold when using other outcomes (e.g., in- and extra-role performance, mental health, anxiety, etc.).

Finally, while our research focused on employee- vs. employer-provided resources (e.g., training), employees and organizations operate in a multilevel context comprising varying industry sectors, governmental economic policies, and national cultural contexts. Therefore, it would also be important to examine the boundary conditions of these relationships across different industry sectors (e.g., high-tech vs. hospitality), economic contexts (e.g., robust vs. weak social safety nets and government-provided retraining investments), and cultural (e.g., high vs. low uncertainty avoidance) contexts.

## 5. Conclusions

Our results indicate that technology-related changes in the workplace (i.e., technology-related changes to information and communication devices, ways of working, and products or services) are associated with increased concerns regarding how such changes will impact the qualitative nature of their job and threaten valued job features (i.e., Qual JI). Employee Qual JI was associated with less job satisfaction, more sleep difficulties, and greater work-to-family conflict. Encouragingly, the adverse effects of technology changes on technology-related Qual JI were buffered when employees received training from their employers, emphasizing the importance of employer-provided resources. On the other hand, the impacts of tech-related Qual JI on work engagement, sleep difficulties, and work-to-family conflict were magnified when employees self-funded their training. Together, these results indicate that organizations (rather than employees) should provide the necessary resources for employees to respond to those technology-induced changes as societies adopt more advanced forms of technology. From a theoretical perspective, our data extend the validity of the JD–R model to conceptualize technology changes as an additional demand faced by employees leading to potential health impairment processes. Future research should test the boundary conditions of these effects across different industries, and sociocultural and economic contexts, as well test the mediating mechanisms using a more rigorous longitudinal design.

## Figures and Tables

**Figure 1 ijerph-19-14368-f001:**
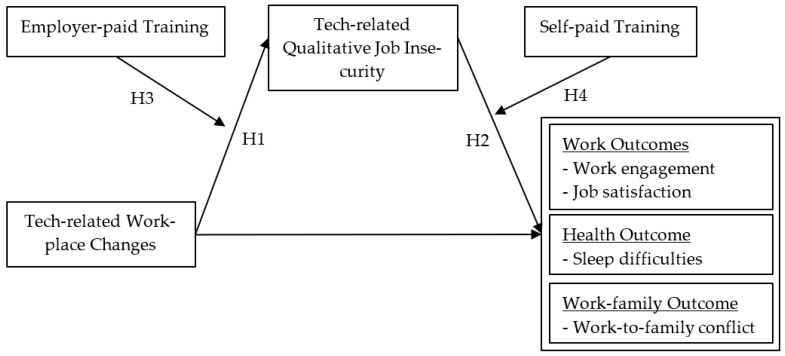
The hypothesized study model.

**Table 1 ijerph-19-14368-t001:** Sociodemographic information of the sample.

Demographics	*N*	%
**Age**		
15–19	144	0.50
20–29	4132	14.30
30–39	6865	23.70
40–49	7761	26.80
50–59	7395	25.50
60 or older	2692	9.30
**Sex**		
Male	13,835	47.72
Female	15,154	52.28
**Education**		
Primary education or lower	363	1.25
Lower secondary education	1161	4.00
Upper secondary education	10,352	35.71
College or above	17,081	58.92
Refused	32	0.11
**Full- vs. Part-time**		
Full-time work	24,936	86.02
Part-time work	3929	13.55
Don’t know/no opinion	116	0.40
Refused	8	0.03
**Permanent vs. Temporary**		
Permanent workers	23,747	81.90
Temporary workers	3761	13.00
**Monthly income**		
Less than 2 million won	7568	26.11
Less than 3 million won	9972	34.40
Less than 4 million won	5975	20.61
More than 4 million won	4242	14.63
Don’t know/no opinion/refused	1232	4.25
**Occupation**		
Administrator	176	0.61
Professional and Semi-professional	6456	22.27
Office worker	7078	24.42
Service worker	3584	12.36
Sales worker	3517	12.13
Agriculture, forestry, and fishery industry skilled worker	124	0.43
Technical skilled worker and related skilled worker	2350	8.11
Equipment machinery operator and assembly worker	2763	9.53
Simple labor worker	2941	10.15

**Table 2 ijerph-19-14368-t002:** Descriptive statistics and correlations of study variables.

	*M*	*SD*	1	2	3	4	5	6	7
1. Tech-Related Changes	0.11	0.28							
2. Tech-Related Qual JI	2.29	0.72	0.49 *						
3. Employer-Paid Training	0.27	0.44	0.22 *	0.00					
4. Self-Paid Training	0.04	0.20	0.11 *	0.00	0.20 *				
5. Work Engagement	3.57	0.67	0.07 *	0.00	0.10 *	0.05 *			
6. Job Satisfaction	2.90	0.50	0.05 *	−0.07 *	0.08 *	0.04 *	0.32 *		
7. Sleep Difficulty	1.64	0.76	0.10 *	0.10 *	0.08 *	0.10 *	−0.17 *	−0.21 *	
8. Work-to-Family Conflict	2.01	0.81	0.11 *	0.16 *	0.06 *	0.09 *	−0.13 *	−0.16 *	0.33 *

*N* = 28,989. Qual JI: Qualitative job insecurity. * *p* < 0.001.

**Table 3 ijerph-19-14368-t003:** Path analysis results for the hypothesized model.

	Outcomes				
	Technology-Related Qual JI	Work Engagement	Job Satisfaction	Sleep Difficulties	Work–to-Family Conflict
**Intercept**	0.007	3.566 **	2.901 **	1.626 **	1.992 **
**Predictors**					
Tech-related changes	0.193 **	0.165 **	0.096 **	0.230 **	0.259 **
Employer-provided training	−0.012				
Tech-related changes × Employer-provided training	−0.125 **				
Tech-related Qual JI		0.004	−0.045 **	0.097 **	0.174 **
Self-paid training		0.127 **	0.077 **	0.340 **	0.342 **
Tech-related Qual JI × Self-paid training		−0.116 **	−0.043	0.118 *	0.110 *
**Residual Variances**	0.522 **	0.442 **	0.246 **	0.558 **	0.635 **

*N* = 28,989. Unstandardized path coefficients are presented. Tech-related changes and Qual JI were mean-centered. Qual JI: Qualitative job insecurity. Training: 0 = No, 1 = Yes. * *p* < 0.01. ** *p* < 0.001.

**Table 4 ijerph-19-14368-t004:** Indirect and total effects of technology-related changes on outcomes via technology-related qualitative job insecurity by training status.

	Training Status					
Outcome	Employer-Provided Training	Self-Paid Training	Estimate	*S.E.*	*t*	*p*-Value
**Indirect effects**						
Work Engagement	No	No	0.001	0.001	0.594	0.552
	No	Yes	−0.022	0.006	−3.420	0.001
	Yes	No	0.000	0.000	−0.583	0.560
	Yes	Yes	−0.008	0.003	−2.342	0.019
Job Satisfaction	No	No	−0.009	0.001	−6.658	<0.001
	No	Yes	−0.017	0.005	−3.199	0.001
	Yes	No	0.001	0.000	2.789	0.005
	Yes	Yes	−0.006	0.003	−2.241	0.025
Sleep Difficulty	No	No	0.019	0.002	7.621	<0.001
	No	Yes	0.041	0.009	4.632	<0.001
	Yes	No	−0.002	0.001	−2.843	0.004
	Yes	Yes	0.015	0.006	2.634	0.008
Work-to-Family Conflict	No	No	0.034	0.004	8.200	<0.001
	No	Yes	0.055	0.010	5.615	<0.001
	Yes	No	−0.004	0.001	−2.870	0.004
	Yes	Yes	0.019	0.007	2.769	0.006
**Total effects**						
Work Engagement	No	No	0.166	0.014	11.998	<0.001
	No	Yes	0.144	0.015	9.423	<0.001
	Yes	No	0.165	0.014	11.966	<0.001
	Yes	Yes	0.158	0.014	11.023	<0.001
Job Satisfaction	No	No	0.088	0.010	8.568	<0.001
	No	Yes	0.079	0.011	6.915	<0.001
	Yes	No	0.097	0.010	9.544	<0.001
	Yes	Yes	0.090	0.011	8.538	<0.001
Sleep Difficulty	No	No	0.293	0.019	15.717	<0.001
	No	Yes	0.314	0.02	15.348	<0.001
	Yes	No	0.255	0.018	13.948	<0.001
	Yes	Yes	0.279	0.019	14.417	<0.001
Work-to-Family Conflict	No	No	0.248	0.018	13.603	<0.001
	No	Yes	0.271	0.02	13.597	<0.001
	Yes	No	0.227	0.018	12.527	<0.001
	Yes	Yes	0.244	0.019	13.01	<0.001

*N* = 28,989. Unstandardized path coefficients are presented.

## Data Availability

The Korean Working Conditions Survey (KWCS) is publicly available at https://www.kosha.or.kr/eoshri/resources/KWCSDownload.do under permission from Korea Occupational Safety and Health Agency (KOSHA).

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
