# Peer review of "Technology Advancements and Employees’ Qualitative Job Insecurity in the Republic of Korea: Does Training Help? Employer-Provided vs. Self-Paid Training"

_ijerph, 2022, doi:10.3390/ijerph192114368_

Round 1

Reviewer 1 Report

Observation figure 1. The hypothesised study model:

·        *  The figure 1 does not contain the hypothesis; accordingly, it is complicated for the reader to understand the relation between variables.

·        *  The implication of risk in the model is not understandable.

Observation per variable in the model

Variable: Teach-related Workplace Changes

Observation: In the theoretical framework, it is not clear what encompasses the “Workplace changes”, may it be dismissals? In hypothesis 3, “Technology-related changes” is used in the wording, does it refer to “Teach-related workplace changes”?

Variable: Employer-paid Training - Self-paid Training

Observation: In the point 1.1, specify with examples the difference between the propositions of paragraph 2. With an example, explain why self-financed training increases the deterioration of health. The text implies that if a worker decides to self-finance master studies, their health will worsen, i.e., they will not be able to sleep well? Is that what it means?

Variable: Health Outcome: Sleep Difficulties

Observation: Why was only considered the difficulty sleeping? In the theoretical framework is explained that other studies determined other factors, why not to consider different emotional disorders (affectations) as anxiety or panic attacks.

Hypothesis:

In hypothesis 2, the wording is confusing and complicated. It seems there is a mixture of hypothesis within this one.

Methodology:

The use of cross-sectional data is specified; nevertheless, the type of quantitative methodology is not exactly stated.

Other sections:

It is not stated if hypotheses are approved or rejected.

Reviewer 2 Report

The authors presented a large study focused on evaluating the associations between technological advancement in the workplace with employee outcomes in the Republic of Korea. 

The manuscript presents new information and is relevant in the context of Health and Safety in the IV Industrial Revolution Era. 

There are some minor aspects that should be addressed before considering it for publication

Since your data is limited to the Republic of Korea, I respectfully suggest modifying the title including it - "Technology Advancements and Employees’ Qualitative Job Insecurity in the Republic of Korea: Does Training Help? Employer-provided vs. Self-paid Training"

Please include the ethics approval. In case you don´t need it due to is a public dataset, please declare it.

The model presented and explained in the introduction is quite relevant. Nevertheless, I reckon it may work better as supplementary material.

Please include a reference related to the 6th Korean Working Conditions Survey (KWCS).

I was wondering if the KWCS has data from 50,000 workers, and why you have included only 28,989 workers. Please explain and consider the possible implications for your results.

Also, please explain who was in charge of the scales translation and the validation process of the questions. 

The first paragraph of the results section should be moved to the methods section.

Table 1. It is not clear to me what the numbers of columns mean. 

Table 3. Why did you present the unstandardized path coefficients? It is possible to report the standardized ones? Or standardized ones correspond to the column "Est./S.E."? Please clarify

Please adjust the references to the journal style.

Please adjust the abstract to the journal guidelines

Also, please consider presenting a table with the sociodemographic information from the 28,989 workers (including, sex, age, socioeconomic information, economic sector, type of work, etc)

Reviewer 3 Report

The Abstract is well prepared, however it could be improved if the authors would include in the Abstract the Research Question, as well as the Research Methodology that the authors followed. This information is revealing and is necessary in the Abstract.

The first paragraph of the Introduction has relevant objective information, and aims to frame the research developed by the authors, such as "AI has grown by 270% in four years, with more than 90% of businesses having...". However, without citations, that is, without indicating the fonts of information, they place many doubts about them. It is recommended that authors place the references throughout the article.

Authors should review the referencing system throughout the article, given that they use the APA format when this is not the one used by MDPI. Authors should consult the guidelines for authors, and see which is the recommended format for publishing in this journal.

The authors note that the sample chosen for this study was 28,989 workers in the Republic of Korea collected in 2021. However, why did the authors choose this sample? What are the variables in this sample that led the authors to choose it? This information is very important to inform the research, and should be justified by the authors.

How did the authors validate the data obtained? Was any statistical analysis/thesis made of the data obtained? It would be very important to ensure the reliability of the data obtained.

The conclusions are very vague and should be reviewed and strengthened by the authors. It would be better if the authors put here some of the most important values obtained with this research, and reinforce in this way the importance and contribution of this research to the society and scientific community.

It would be very important that the authors at the end of the conclusions indicate the limitations of this research. All research has its limitations, and they are a natural part of research. However, the authors' sharing of their limitations helps the scientific community to understand the research and some of the results obtained.

Round 2

Reviewer 3 Report

The authors have strongly improved the article, and made the corrections requested by the reviewers!